# Ferric Ions Crosslinked Epoxidized Natural Rubber Filled with Carbon Nanotubes and Conductive Carbon Black Hybrid Fillers

**DOI:** 10.3390/polym14204392

**Published:** 2022-10-18

**Authors:** Kriengsak Damampai, Skulrat Pichaiyut, Klaus Werner Stöckelhuber, Amit Das, Charoen Nakason

**Affiliations:** 1Department of Rubber Technology, Faculty of Science & Industrial Technology, Prince of Songkla University, Surat Thani 84000, Thailand; 2Leibniz-Institut für Polymerforschung Dresden e.V., D-01069 Dresden, Germany

**Keywords:** epoxidized natural rubber, ferric chloride, carbon nanotubes, nanocomposite, conductive carbon black

## Abstract

Natural rubber with 50 mol % epoxidation (ENR-50) was filled with carbon nanotubes (CNTs) and conductive carbon black (CCB) hybrid fillers with various CCB loadings of 2.5, 5.0, 7.0, 10.0 and 15.0 phr, and the compounds were mixed with ferric ion (Fe^3+^) as a crosslinking agent. The ENRs filled exclusively with CNTs, and CNT–CCB hybrid fillers exhibited typical curing curves at different CCB loadings, i.e., increasing torque with time and thus crosslinked networks. Furthermore, the incorporation of CNT–CCB hybrid fillers and increasing CCB loadings caused an enhancement of tensile properties (modulus and tensile strength) and crosslink densities, which are indicated by the increasing torque difference and the crosslink densities. The crosslink densities are determined by swelling and temperature scanning stress relaxation (TSSR). Increasing CCB loadings also caused a significant improvement in bound rubber content, filler–rubber interactions, thermal resistance, glass transition temperature (*T_g_*) and electrical conductivity. A combination of 7 phr CNT and CCB with loading higher than 2.5 phr gave superior properties to ENR vulcanizates. Furthermore, the secondary CCB filler contributes to the improvement of CNT dispersion in the ENR matrix by networking the CNT capsules and forming CNT–CCB–CNT pathways and thus strong CNT–CCB networks, indicating the improvement in the tensile properties, bound rubber content and dynamic properties of the ENR composites. Moreover, higher electrical conductivity with a comparatively low percolation threshold of the hybrid composites was found as compared to the ENR filled with CNTs without CCB composite. The superior mechanical and other properties are due to the finer dispersion and even distribution of CNT–CCB hybrid fillers in the ENR matrix.

## 1. Introduction

Epoxidized natural rubber (ENR) can be chemically modified by a reaction with peracid to develop epoxirane rings onto NR molecular structures [1,2]. The ENR gains a number of superior properties, such as low gas permeation, high polarity, high compatibility with polar polymers and high electrical conductivity, which cannot be realized by unmodified NR [3]. ENR is more or less a biodegradable biopolymer which is often used to prepare a variety of rubber composites [4]. In addition, ENR has considerable contents of polar epoxy groups and reactive carbon–carbon double bonds in its molecular structure, enabling it to react with polar materials and metal ions [5]. These properties lead to a possibility of introducing functional groups and/or newly formed compounds between ENR molecules, thereby improving important properties of rubber vulcanizates, including dynamic and mechanical properties.

A number of research works have been conducted to reinforce ENR compounds by introducing different types of fillers and curing systems. That is, ENR has been vulcanized by different vulcanization systems, including sulfur [6,7,8], peroxide [9,10], dicarboxylic curing systems [11] and imidazole [12]. Furthermore, it was found that some types of metal ions could react with ENR molecules via the reactive oxirane rings, leading to a crosslinking reaction of ENR molecules by forming the permanent coordination crosslinked network structures [13,14]. In epoxy resins, it was found that the metal ions chromium (III), titanium (IV) oxyacetylacetonate, zirconium (IV), cobalt (II) and cobalt (III) acetylacetonates are particularly reactive with anhydride-cured epoxy resins by adding these chemicals to the resin at a loading range of 0.05–0.10 wt %. It was also reported that the reactions had fast gel times associated with good storage stabilities at room temperature [15]. Furthermore, metal acetylacetonates are effective latent catalysts for epoxy/anhydride systems. That is, the metal complexes have potential to act as the curing agents in malemide epoxy resins [14]. In ENR, the ring opening reaction of ENR with 50 mol % (i.e., ENR-50) was conducted in the presence of SnCl_2_∙2H_2_O catalyst. The ENR/Sn complex hybrid was thereafter formed via insertion of SnCl_2_ into the quaternary and methine carbon of ring-opened ENR molecules under CO_2_ atmosphere [16]. Simultaneously, it was found that the ferric ion (Fe^3+^) is one the most efficient metal ions to promote the crosslinking reaction of ENR molecules [17]. This means that the opened ring products of ENR are able to form new compounds with other ring-opened ENR fragments via Fe^3+^ bridges through coordination bonds to link the ENR molecular chains together [17]. ENR molecules containing an epoxide group can even undergo an internal polymerization reaction and yield many exchanged and complicated polymeric microstructures, which can lead to strong crosslinking structures in ENR molecules, as described in our previous work [17,18]. Furthermore, oxygen-rich commercial rubbers such as ENR could be used to create metal–oxygen coordination crosslinking complexes, which have been shown to be highly effective in improving the mechanical performance of rubber [19]. Self-healable ENR molecular networks have been prepared by reacting ENR with mixed diamine and metal ions, [iron (III) chloride] crosslinking agents. It was found that the dynamic properties of the metal ion crosslinked rubber vulcanizates are comparable with the conventional sulfur-vulcanized samples [14].

Epoxidized natural rubber (ENR) has been reinforced by various fillers, including montmorillonite clay [20], conductive carbon black (CCB) [21], geopolymers [22] and carbon nanotubes (CNTs) [23]. Carbon nanotubes were first discovered in 1991 [24] and can enhance several rubber composite properties, including mechanical properties, electrical conductivity and thermal stability [23]. However, the mechanical properties of the developed rubber composites typically depend on the quality of the dispersion of filler in the rubber matrix. It is noted that CNTs have a high aspect ratio and strong van der Waals forces, causing strong agglomeration in the rubber matrix and hence inferior mechanical properties [24,25,26]. Thus, incorporation of a second filler into the rubber–CNT composite system may improve the CNTs’ dispersion and distribution and enhance the mechanical properties, thermal stability, electrical conductivity and other related properties [27,28,29]. Various types of hybrid fillers have been investigated in polymer systems containing CNT-filled ENR and epoxy, including carbon black [30,31], nano-clay (NC) [32], graphene nano platelets [33], zinc oxide [34], silver nanoparticles (AgNPs) [35] and conductive carbon black (CCB) [36]. It was found that the secondary filler generally contributes to improve the CNTs’ dispersion in the rubber matrix [36]. Moreover, they could connect to the CNT encapsulates and form conductive CNT–CCB–CNT pathways in the ENR composites, causing significant improvement in the electrical conductivity [36]. The novelty of the current work is that we are the first to study the influence of CNT–CCB hybrid fillers for the reinforcement of epoxidized natural rubber (ENR) crosslinked by coordination bonds via Fe^3+^ linkages. Furthermore, CCB in the hybrid filler system provides a new filler network of CNTs by connecting the CNT bundles, promoting superior electrical and mechanical properties of ENR nanocomposites.

In this work, epoxidized natural rubber (ENR) was crosslinked after compounding with Fe^3+^ from ferric chloride (FeCl_3_) and then reinforced with a hybrid filler system that consists of carbon nanotubes (CNTs) and conductive carbon black (CCB). The main aim was to increase filler–rubber interactions between the different filler particles and the rubber matrix in order to form a three-dimensional filler network with high electrical conductivities and tensile properties. Furthermore, cure characteristics, mechanical properties, crosslink densities, bound rubber content, temperature scanning stress relaxation (TSSR), dynamic properties, thermal properties and electrical conductivity were investigated.

## 2. Materials and Methods

### 2.1. Materials

Epoxidized natural rubber with epoxide content of 50 mol % (ENR-50) was manufactured by Muangmai Guthrie Co. Ltd. (Surat Thani, Thailand). Ferric chloride (FeCl_3_) was manufactured by Sigma-Aldrich Pte Ltd., Darmstadt, Germany. The multi-wall carbon nanotubes (CNTs, NC7000) with average dimeter of 9.5 nm, average length of 1.5 µm and 90% purity were produced by Nanocyl S.A. (Sambreville, Belgium). The conductive carbon black (CCB), Vulcan XC72, with a particle diameter of 30 nm and density at 20 °C of 1.7–1.9 g/cm^3^ was supplied by Cabot Corporation, TX, USA. Other chemical components used in the compounding formulation are summarized in Table 1.

### 2.2. Preparation of ENR–FeCl_3_/CNT–CCB Hybrid Nanocomposites

The ENR-50 was first compounded with an optimum loading dose of FeCl_3_ at 7 mmol and 7 phr of CNTs, according to our previous work [17]. The mixing was performed by an internal mixer, Brabender Plasticorder with Mixer 50 EHT model 835205 (Duisburg, Germany), at 60 °C and a rotor speed of 60 rpm, according to the formulation shown in Table 1. The ENR-50 was first masticated for about 2 min, and then 7 mmol FeCl_3_ was added and continued mixing for another 2 min. Then, 7 phr of CNTs was incorporated and mixed for 6 min to reach the total mixing time of 10 min. This sample is designated as “F7-CNT7”. In the preparation of ENR filled with CNT–CCB hybrid filler compounds, ENR-50 was also firstly masticated and mixed with 7 mmol FeCl_3_ and 7 phr of CNTs, similar to the preparation of F7-CNT7, but mixing was conducted for about 3 min after incorporating CNTs before adding each loading of CCB at 2.5, 5, 7, 10 and 15 phr. After that, the rubber compounds continued mixing for another 3 min to reach the total mixing time of 10 min. The rubber samples are designated as “CNT7/CCB_X_”, where x is the CCB loading in phr. The compound (i.e., ENR–FeCl_3_/CNT–CCB hybrid nanocomposite) was then dumped and sheeted out on a two-roll mill, model YFCR 600, Yong Fong machinery Co., Ltd. (Samut Sakorn, Thailand) at ambient temperature. This was aimed to enhance the dispersion and distribution of the fillers in the ENR matrix. The rubber compounds were eventually cured in a standard hot press under high pressure and temperature conditions of 1500 psi at 160 °C.

Cure characteristics of rubber compounds were investigated by a moving die rheometer (MDR), model 81001 MDR 2000, Alpha Technologies (Hudson, OH, USA), at 160 °C. A heated press (PR1D-W400L450PM compression molding machine, Charoen Tut Co., Ltd., Samut Sakorn, Thailand) was used to prepare the ENR-50 vulcanizates at 160 °C using the respective cure (*t_c_*_90_) time based on the rheometer test. The compounding and characterization procedures of ENR–FeCl_3_/CNT–CCB hybrid nanocomposites are summarized and described in Figure 1.

### 2.3. Cure Characterization

The cure characteristics of the rubber compounds were measured by means of a moving die rheometer (MDR 2000, Alpha Technologies, OH, USA) to investigate the curing behavior of rubber compounds at 160 °C with the strain amplitude of 1° arc and a fixed frequency of 1.67 Hz. The optimum scorch time (*t_s_*_1_), cure time (*t_c_*_90_), minimum torque (*M_L_*), maximum torque (*M_H_*) and torque difference (*M_H_*-*M_L_*) were determined from the curing curves.

### 2.4. Tensile Properties

The tensile test specimens were prepared by die cutting from the vulcanized rubber sheet to form a dumbbell-shaped specimen, according to ISO 527 (type 5A). The test was then performed using a tensile testing machine, Zwick GmbH & Co., KG (Ulm, Germany), at room temperature with a crosshead speed of 200 mm/min, according to ISO 527.

### 2.5. Morphological Characterization

The morphological properties of the CNT- and CNT–CCB-filled ENR–FeCl_3_ compounds were characterized by scanning electron microscopy (SEM), Quanta 250 (Černovice, Czech Republic). Each specimen was cryogenically cracked in liquid nitrogen to create a fresh cross-sectional surface. Then, the dried surface was gold-coated and examined by SEM.

### 2.6. Temperature Scanning Stress Relaxation (TSSR)

Temperature scanning stress relaxation (TSSR) experiments were performed by using a TSSR meter (Brabender*^®^* GmbH & Co. KG, Duisburg, Germany). Dumbbell-shaped specimens were first cut from the vulcanized rubber sheet with die type 5A, according to ISO 527. Before testing, the samples were first annealed in a hot air oven at 100 °C for 30 min to remove thermal history and the intrinsic storage hardening effect of natural rubber molecules [37]. The samples were then cooled to room temperature for about 30 min before placing in the sample holder of the TSSR machine. Then, the rubber sample was stretched to 50% elongation as compared to its original length at 23 °C and thereafter conditioned at this isothermal condition for about 2 h. After that, the non-isothermal experiments were executed by increasing the temperature from 23 °C to 220 °C with a constant heating rate at 2 °K/min until the stress relaxation was complete. From the resulting stress–temperature or force–temperature curves, the relaxation spectrum (*H*(*T*)) was also derived using the relationship between relaxation modulus (*E*(*T*)) and temperature (*T*), as follows [37]:(1)$$H\left(T\right)=-T{\left[\frac{dE\left(T\right)}{dT}\right]}_{\nu =const}$$

### 2.7. Crosslink Density

Crosslink densities of ENR nanocomposites were determined by two different approaches: equilibrium swelling measurement and the temperature scanning stress relaxation (TSSR) test. In equilibrium swelling measurements, the tests were carried out to determine the crosslink density in the rectangular 10 × 10 × 2 mm^3^ rubber specimens. The samples were first weighed before immersing into toluene at room temperature for seven days. The swollen rubber samples were then removed and excess liquid on the specimen surfaces was removed by blotting with filter paper. After that, the specimens were dried in a vacuum oven at 40 °C for 24 h. Finally, the swollen samples were weighed, dried and weighed again. The crosslink density of the rubber vulcanizates was eventually determined according to the Flory–Rehner relation [38]:(2)v=−In1−∅p+∅p+x · ∅p2VL · (∅p13−∅p2)
where ∅p is the volume fraction of rubber in the swollen network, *V_L_* is the molar volume of toluene and *x* is the interaction parameter of polymer and solvent (for ENR and toluene, the value is 0.34) [37].

Crosslink density of the ENR vulcanizates was also estimated by the temperature scanning stress relaxation (TSSR) test. Typically, the apparent crosslink density can be estimated from the maximum slope in the initial part of the stress–temperature curve (TSSR), using Equations (4) and (5), as described in detail elsewhere [34].
(3)$$V_{e}=\frac{k}{R^{*}\left(\lambda -\lambda^{2}\right)}$$
(4)$$V_{e}=\frac{\rho }{M_{C}}$$
where $V_{e}$ is the apparent crosslink density (mol/m^3^), $R^{*}$ is the universal gas constant, *λ* is the nominal strain ratio, k is the temperature coefficient of stress (the derivative of mechanical stress with respect to temperature), ρ is the mass density and $M_{C}$ is defined as the average molar mass of the elastically active network chains [39,40].

### 2.8. Bound Rubber Contents

Bound rubber content (BRC) of the ENR–FeCl_3_/CNT–CCB hybrid nanocomposites was estimated by the swelling method [39]. First, the rubber compound (approximately 0.2 g) was cut into small pieces and placed into a stainless steel cage of known weight. The cage was then immersed in 20 mL of toluene for 3 days at room temperature, and the toluene was renewed every 24 h. After 3 days, the rubber and cage were removed from the solvent and then dried in a hot air oven at 105 °C for 24 h. The sample was again immersed in toluene and left for about 3 days. Finally, the sample was removed and dried at 105 °C for about 24 h and weighed. The bound rubber content was eventually calculated by the following equation [41]:(5)$$Bound\;rubber\;content\;\left[\%\right]=\frac{W_{fg}-W_{f}}{W_{p}}$$
where *W_f_* and *W_p_* are the weights of filler and rubber in the specimens, respectively. *W_fg_* is the weight of filler with bound rubber absorbed on it after toluene extraction [41].

### 2.9. Payne Effect

The Payne effect of rubber compounds was determined by measuring the storage modulus as a function of double strain amplitude by using a rubber process analyzer (RPA) (Alpha Technologies, Akron, OH, USA). The strain sweep analysis was conducted in the strain ranges of 0–100% with an oscillating frequency of 1 Hz. The test temperature was 100 °C. The Payne effects of various ENR compounds were estimated by the following equation [42]:(6)$$\%\;\mathrm{Payne}\;\mathrm{effect}\;=\frac{G\prime_{max-}G\prime_{min}\;}{G\prime_{min}}\times 100$$
where *G′_max_* and *G′_min_* are the maximum and minimum of storage moduli (*G*′).

### 2.10. Dynamic Mechanical Analysis

Dynamic mechanical properties were obtained from a Perkin Elmer Dynamic Mechanical Analyzer (DMA 8000) (Perkin Elmer Inc., Waltham, MA, USA). The experiments were carried out in a tension mode within the temperature ranges from −100 °C to 100 °C with a heating rate of 3 °C/min at a fixed deformation frequency of 1 Hz.

### 2.11. Electrical Properties

Electrical properties such as electrical conductivity (*σ*) and dielectric constant ($\varepsilon^{\prime }$) of the ENR–FeCl_3_ compound and ENR–FeCl_3_/CNT–CCB hybrid nanocomposites were measured at room temperature by an LCR meter (Hioki IM 3533, Hioki E.E. Corporation, Nagano, Japan) in the frequency ranges of 1 to 10^5^ Hz. The LCR meter was connected to the electrode plates of a dielectric test fixture model 16451B dielectric test fixture (Test Equipment Solutions Ltd., Berkshire, UK) with a 5 mm electrode diameter. The electrical conductivity (*σ*) and dielectric constant (*ε’*) were determined as follows [43]:(7)$$\sigma =\frac{1}{\rho }=\frac{d}{\left(R_{p}\right)A}$$
(8)$$\varepsilon^{\prime }=\frac{C_{p}\left(d\right)}{A\left(\varepsilon_{0}\right)}$$
where *d* is the thickness of the specimen placed between two electrodes, *A* refers to the area of an electrode, *R_p_* is resistance and *C_p_* is capacitance. The parameter $\varepsilon^{\prime }$ is the dielectric constant of the free space, which is 8.854 × 10^−12^ F/m. The factor ρ is the volume resistivity, which is the reciprocal of conductivity.

## 3. Results and Discussion

### 3.1. Curing Characteristics

Figure 2 shows the cure curves of ENR filled with CNTs (without FeCl_3_), ENR filled with CNT–CCB hybrid filler (without FeCl_3_), ENR–FeCl_3_ filled with CNTs without CCB (F7-CNT7) and with 7 phr CNTs in combination with various CCB loadings at 2.5, 5.0, 7.0, 10.0 and 15.0 phr (i.e., CNT7/CCB2.5, CNT7/CCB5.0, CNT7/CCB7.0, CNT7/CCB10.0 and CNT7/CCB15.0). It can be seen that the ENR-50 (without FeCl_3_) filled with CNTs and CNT–CCB hybrid filler had no response in the cure curve. This is attributed to the absence of a crosslinking reaction between ENR molecules. However, a curing reaction of ENR-50 with 7 mmol FeCl_3_ (ENR–FeCl_3_) filled with 7 phr CNTs (F7-CNT7) is observed from increasing torque with time. This led to the chemical reaction between oxirane rings in ENR molecules and caused Fe^3+^ ion to form rubber network structures via –O–Fe–O– coordination linkages [18]. Moreover, the internal polymerization from the reaction of ENR with epoxy groups could take place, resulting in many complicated polymeric microstructures, a mixture of isomeric and homologous compounds [44]. This results in strong crosslinking structures in ENR molecules [17,18]. Furthermore, the ENR–FeCl_3_ filled with 7 phr of CNTs and CNT–CCB hybrid fillers with various CCB loadings show increased mixing torque–time curves with increasing CCB loadings. The matching curing curves, i.e., increasing torque with time, are clearly seen in Figure 2. This may be because the crosslinking reaction might require a longer time to complete its reaction to approach the equilibrium state.

Table 2 shows the curing data of the ENR compounds. The values of minimum torque (*M_L_*), maximum torque (*M_H_*), torque difference (*M_H_-M_L_*), scorch time (*t_s_*_2_) and cure time (*t_c_*_90_) are extracted from the rheometric curing curves. I maximum torque and torque difference of the ENR–FeCl_3_ filled with CNTs (i.e., F7-CNT7) had the lowest values, but these properties were higher with incorporating and increasing CCB loadings. This is due to a finer filler dispersion and distribution in the ENR matrix. That is, the CCB particles might hinder the CNTs’ agglomeration. Therefore, the CCB secondary filler contributes to improving the CNTs’ dispersion and distribution in the rubber matrix by connecting to the CNT encapsulates and forming CNT–CCB–CNT pathways in the ENR matrix, causing significant improvements in the maximum torque and torque difference [36].

In Table 2, it is also seen that the scorch time (*t_s_*_2_) and cure time (*t_C_*_90_) of ENR compounds were shortened by the addition of CCB. Moreover, the *t_s_*_2_ and *t_C_*_90_ decreased with increasing CCB loadings. Furthermore, incorporating and increasing CCB loadings caused an increasing cure rate index (CRI). It is noted that the cure rate index (CRI) is a measurement of the cure rate based on the differences between the optimum cure time (*t*_90_) and incipient scorch time (*t_s_*_2_). This is due to higher thermal conductivity of CCB and CNTs that cause enhancing and accelerating of the curing reaction of the ENR compounds [31]. Furthermore, increasing the CCB loading causes an increase in thermal conductivity of the ENR composites that may facilitate and hence accelerate the curing reaction. This also increases the chemical interaction between polar functional groups in ENR molecules and filler surfaces (CNTs and CCB) [45].

Higher contents of CCB in the ENR composite also cause an acceleration of the crosslinking reaction between the Fe^3+^ and ENR molecules via oxirane groups due to higher thermal conductivity. The ENR-50 with CNT–CCB hybrid filler had a higher torque difference than the ENR with only CNTs. This indicates higher reinforcement due to higher crosslink density and higher solid particulate contents. This indicates higher chemical linkages between ENR molecules via Fe^3+^ and also stronger chemical interactions between polar functional groups in ENR molecules and CNT–CCB hybrid fillers’ surfaces.

### 3.2. Tensile Properties

Figure 3 shows stress–strain curves of ENR-50 with 7 mmol of FeCl_3_ (FeCl_3_-ENR), filled with 7 phr of CNTs (F7-CNT7) and CNT–CCB hybrid fillers with various CCB loadings at 2.5, 5.0, 7.0, 10.0 and 15.0 phr (i.e., CNT7/CCB2.5, CNT7/CCB5.0, CNT7/CCB7.0, CNT7/CCB10.0 and CNT7/CCB15.0, respectively). Tensile properties in terms of 100% modulus, tensile strength and elongation at break of various ENR compounds are summarized in Table 3. It is clearly seen that adding and increasing CCB loadings affect the characteristics of the stress–strain behaviors of ENR compounds by significantly increasing the 100% modulus and tensile strength as compared with the ENR filled with only CNTs (F7-CNT7). This is due to the reinforcing efficiencies of CNT–CCB hybrid fillers because the strong CNT–CCB networks were formed in the ENR matrix. Moreover, the finer filler dispersion and distribution due to the newly formed CNT–CCB networks strongly affected the enhancement of the rubber–filler interaction.

In Table 3, increasing CCB loadings in CNT–CCB hybrid fillers also caused an increase in the 100% modulus and tensile strength. This is again due to the formation of the stronger CNT–CCB filler networks dispersed in the ENR matrix together with stronger crosslinking ENR networks via the Fe-O-Fe coordination bridges and the internal polymerization. In Figure 3 and Table 3, it is also clearly seen that the elongation at break decreased upon increasing CCB loadings. This is normally observed in rubber composites with well-dispersed reinforcing fillers that restrict the chain mobility of rubber molecules by rubber–filler interactions.

### 3.3. Morphological Properties

Figure 4 shows SEM micrographs of ENR–FeCl_3_ filled with CNTs (F7-CNT7) and CNT–CCB hybrid filler with various CCB loadings at 0, 2.5, 5.0, 7.0, 10.0 and 15.0 phr. The F7-CNT7 shows uneven dispersion of CNT bundles and some agglomerates in the ENR matrix (Figure 4a). Furthermore, in the ENR-50 with CNT–CCB hybrid fillers, the CCB particles are partially connected to form more dispersive CNT networks in the ENR matrix, as seen in Figure 4b–e. Therefore, the CCB particles and their small aggregates may act as the filler bridges at the end of the CNT bundle to form stronger and more dispersive filler networks in the ENR matrix. However, in Figure 4f, the large filler agglomerates are seen in the ENR-50 filled with CNT–CCB hybrid filler at a CCB loading of 15 phr due to the excess amount of CCB particles. Despite the large agglomerates of CCB, their dispersion and distribution in the ENR matrix still provide favorable tensile properties and other related properties of ENR composites.

### 3.4. Payne Effect

Figure 5 shows storage modulus as a function of strain amplitude of ENR–FeCl_3_ filled with CNTs (F7-CNT7) and CNT–CCB hybrid fillers with various CCB loadings at 2.5, 5.0, 7.0, 10.0 and 15.0 phr. It can be clearly seen that the storage modulus of the F7-CNT7 compound shows a constant storage modulus in the low strain region (i.e., lower than 20% strain), but a slight decreasing trend is seen when the strain amplitude is higher than 20%. This may be due to the breakdown of CNT filler networks, indicating the level of the filler–filler or CNT–CNT interaction in the ENR matrix. Furthermore, the storage moduli of the ENR–FeCl_3_ filled with CNT–CCB hybrid fillers were dramatically decreased after strain amplitudes higher than 20% due to the breakdown of the hybrid filler network structures.

Table 4 shows the Payne effect of ENR–FeCl_3_ filled with CNTs (F7-CNT7) and CNT–CCB hybrid fillers with various loadings of CCB. It is noted that the Payne effect can lead to deformation-induced changes in the material’s microstructure, i.e., to breakage and recovery of weak physical bonds linking adjacent filler clusters. Therefore, in Table 4, it is clearly seen that the Payne effect or filler–filler interaction increases with increasing CCB loadings. In Figure 5, it is also seen that the storage modulus of the ENR–FeCl_3_ filled with CNT–CCB hybrid filler with CCB loadings higher than 7 phr showed slightly different modulus–strain curves as compared to the composites with CCB loadings lower than 5 phr. This might be attributed to the large agglomeration of fillers in the ENR matrix for the composites with CCB loadings higher than 7 phr (Figure 4e,f).

### 3.5. Bound Rubber Contents

Figure 6 shows the bound rubber contents of ENR–FeCl_3_ filled with 7 phr of CNTs (F7-CNT7) and CNT–CCB hybrid fillers with various concentrations of CCB at 2.5, 5.0, 7.0, 10.0 and 15.0 phr (i.e., CNT7/CCB2.5, CNT7/CCB5.0, CNT7/CCB7.0, CNT7/CCB10.0 and CNT7/CCB15.0, respectively). It is seen that F7-CNT7 filled with only CNTs displays the lowest bound rubber content. Moreover, the bound rubber content of the ENR filled with the CNT–CCB hybrid composite is higher than the ENR–CNT (F7-CNT7) composite, and it increases with an increase in CCB loading. This is attributed to an increase in filler–filler interaction (i.e., Payne effect in Table 4) and filler–rubber interactions between the polar functional groups of ENR molecules and the polar groups at the filler particle surface (i.e., CNTs and CCB). This is also due to the finer dispersion and distribution of hybrid filler networks in the ENR matrix (Figure 4), which caused increasing filler–rubber interactions, as indicated by the higher storage moduli in Figure 5.

### 3.6. Temperature Scanning Stress Relaxation (TSSR)

The relaxation modulus of ENR–FeCl_3_ filled with 7 phr of CNTs (F7-CNT7) and CNT–CCB with various CCB loadings at 2.5, 5.0, 7.0, 10.0 and 15.0 phr as a function of temperature can be found from Figure 7. The ENR–FeCl_3_ filled with CNT–CCB hybrid fillers showed a higher initial modulus and modulus at a given temperature than the F7-CNT7 without CCB. Moreover, these properties increased with increasing CCB loadings. This might be due to the higher level of crosslink structures, bound rubber content (Figure 6), filler–filler interactions and filler–rubber interactions (Figure 5). This causes an enhancement of the F7-CNT7/CCB*_X_*, as compared with the F7-CNT7 without CCB, which also correlates well to higher torque difference (Table 2), 100% modulus (Table 3) and crosslink density (Table 5). It is noted that the crosslink density of the ENR vulcanizates was also calculated from the maximum slope of the outset part of the stress–temperature curve (Figure 7), using Equations (3) and (4) [39], as described in the following section.

### 3.7. Crosslink Density

Table 5 shows the crosslink densities of ENR–FeCl_3_ and its filled composite with 7 phr of CNTs (F7-CNT7) and CNT–CCB hybrid fillers with various CCB loadings at 2.5, 5.0, 7.0, 10.0 and 15.0 phr. There are two different approaches to estimate the crosslink densities of ENR vulcanizates in this work: the swelling measurement via the Flory–Rehner relation (Equation (2)), and the TSSR measurement based on the maximum slope in the initial part of the stress–temperature curve (Equations (3) and (4)). In Table 5, it is clearly seen that the resulting crosslink densities from the TSSR evaluation and swelling measurements show large differences in magnitude but have the same trends [38]. That is, the unfilled ENR–FeCl_3_ compound shows the lowest crosslink density. The addition of CNTs and CNT–CCB hybrid fillers led to an increase in crosslink densities, which are in agreement with higher torque difference (*M_H_*-*M_L_*) (Table 2) and bound rubber contents (Figure 6) of the ENR-50 with filler and increasing CCB loadings. In Table 5, the highest crosslink density was observed in the ENR–FeCl_3_ filled with the CNT–CCB hybrid filler with a CCB loading of 15 phr. This may be attributed to the highest level of chemical interaction between polar functional groups on the CNT–CCB surfaces and the ENR molecular networks that bridge by the coordination –Fe–O–Fe– linkages.

### 3.8. Dynamic Mechanical Analysis (DMA)

Figure 8 shows the storage modulus (E’) and loss tangent (tan δ) as functions of temperature of ENR–FeCl_3_ filled with 7 phr of CNTs (F7-CNT7) and CNT–CCB hybrid fillers with various CCB loadings at 2.5, 5.0, 7.0, 10.0 and 15.0 phr. The ENR–FeCl_3_ filled with CNT–CCB hybrid fillers had higher storage moduli in the glassy region (i.e., in the temperature ranges from −60 °C to −30 °C) than the ENR filled only with CNTs (F7-CNT7). Moreover, the storage moduli increased with an increase in CCB loading. This may be due to the higher reinforcement of ENR by filler networks of CNT bundles and CCB particles, resulting in stronger CNT–CCB networks dispersed in the ENR matrix. Additionally, more CCB solid contents in the ENR nanocomposites cause higher stiffness in materials. In Figure 8, it is also seen that the addition of CNT–CCB hybrid fillers caused shifts in the tan δ peaks (Figure 8B) and hence the glass transition temperature (*T_g_*) (Table 6) toward higher temperature ranges as compared with the ENR filled only with CNTs (F7-CNT7). This resulted in an increase in *T_g_* of ENR–FeCl_3_ filled with CNT–CCB hybrid filler with increasing CCB loadings. This is attributed to more rigidity of the ENR molecular networks with an increase in CCB loading, due to higher interaction among polar functional groups at CNT–CCB surfaces and the polar groups in ENR molecules, resulting in less chain flexibility and hence higher glass transition temperature. In Figure 8B, it can also be seen that the height and area underneath the transition peak of the ENR–FeCl_3_ filled with CNT–CCB hybrid fillers decreased with an increase in CCB loading with an abruptly decreasing trend in the composites with CCB loadings at 10 and 15 phr. This might be due to the large filler agglomeration in the ENR matrix, as evidenced in SEM micrographs in Figure 4.

### 3.9. Electrical Properties

The dependence of the electrical conductivity on the frequency at room temperature of ENR–FeCl_3_ filled with CNTs (F7-CNT7) and CNT–CCB hybrid fillers with various CCB loadings at 2.5, 5.0, 7.0, 10.0 and 15.0 phr is shown in Figure 9. It can be seen that the electrical conductivity of the F7-CNT7 strongly increased with increasing frequencies or frequency-dependent electrical conductivity. However, in the ENR–FeCl_3_ filled with CNT–CCB hybrid fillers, less frequency-dependent curves are seen, with a marginal increase in electrical conductivity with increasing frequency. This may be due to more free electron movement in the ENR matrix. Furthermore, the ENR filled with CNT–CCB hybrid composites shows higher electrical conductivity than the ENR–CNT composite. In addition, the electrical conductivity at a given frequency increases with increasing CCB loadings. This is attributed to the formation of more conductive CNT–CCB networks in the ENR matrix. That is, the CCB particles or small CCB aggregates may act as the filler bridges to connect the CNT bundles to form the strong filler networks which are dispersed in the ENR matrix to facilitate the movement of free electrons. This also causes higher filler–filler (CNT–CCB) and rubber–filler interactions between polar functional groups of hybrid fillers and the ENR matrix. This causes the formation of three-dimensional filler networks in the rubber networks with the enhancement of end-to-end electron hopping among CNT–CCB bridges, causing electron transfer and significantly enhancing the electrical conductivity of the rubber composites.

The formation of conductive paths of filler networks is verified by the percolation threshold concentration (CTC) based on the plots of electrical conductivity versus filler concentration, as shown in Figure 10. It is clearly seen that the CTC of the ENR–FeCl_3_/CNT–CCB hybrid composites is lower than 2.5 phr of CCB, where the material turns from an insulator to the conductive rubber material.

Figure 11 shows dielectric constant as a function of frequency of ENR–FeCl_3_ filled with CNTs (F7-CNT7) and CNT–CBB hybrid fillers with various CCB loadings at 2.5, 5.0, 7.0, 10.0 and 15.0 phr. It is clearly seen that the F7-CNT7 has the lowest dielectric constant, and this property is more or less independent of frequency. On the other hand, the ENR filled with CNT–CCB hybrid fillers with various CCB loadings has a higher dielectric constant than the F7-CNT7, but the frequency-dependent dielectric constant is seen. This means the electric current moves through the conductive filler pathways induced by polarization in the ENR vulcanizates [46]. It was also found that an increasing dielectric constant is seen with an increase in CCB loading, which corresponds to the trend of electrical conductivity (Figure 9). In addition, a significantly increasing trend of the dielectric constant of the ENR filled with CNT–CCB hybrid fillers is seen with an increase in CCB loading. This might be due to the increasing polarizability of the fillers and also to anomalous diffusion within aggregates of CNTs and CCB, together with their dispersion in the rubber matrix [47].

## 4. Conclusions

Epoxidized natural rubber with 50 mol % (ENR-50) was successfully crosslinked by Fe^3+^ ion (7 mmol FeCl_3_) to form the coordination linkages (–O–Fe–O–) between oxirane rings in ENR molecules and Fe^3+^ ion. This novel elastomer material was filled with 7 phr of CNTs (F7-CNT7) and CNT–CCB hybrid fillers with various CCB loadings at 2.5, 5.0, 7.0, 10.0 and 15.0 phr. It was found the ENR–FeCl_3_ compound filled with CNTs and the CNT–CCB hybrid filler had shorter scorch and cure times and a higher cure rate index than the ENR–FeCl_3_ filled with only CNTs. Furthermore, the ENR/CNT–CCB hybrid composites indicated higher 100% modulus, tensile strength and crosslink density than the ENR–CNTs composite. This is attributed to a higher level of chemical interaction between polar functional groups in the CNTs, CCB surfaces and the ENR molecular networks. Furthermore, CCB secondary filler caused an improvement in the CNTs’ dispersion in the ENR matrix by connecting to the CNT encapsulates, forming CNT–CCB–CNT pathways and hence strong CNT–CCB networks. This significantly improved the mechanical properties, bound rubber, crosslink density and electrical properties, revealed by the SEM micrograph of the ENR composites with suitable dispersion and distribution of fillers in the rubber matrix. Moreover, the ENR–FeCl_3_ filled with CNTs and the CNT–CCB hybrid filler indicated superior electrical conductivities as compared to the ENR–FeCl_3_ filled with only CNTs. This is also attributed to the formation of stronger conductive CNTs and CCB networks in the ENR matrix. Finally, the novel ENR/CNT–CCB hybrid nanocomposites with coordination crosslinking system and adequate performance in terms of mechanical and electrical properties will encourage further research in the field of smart materials.

## Figures and Tables

**Figure 1 polymers-14-04392-f001:**
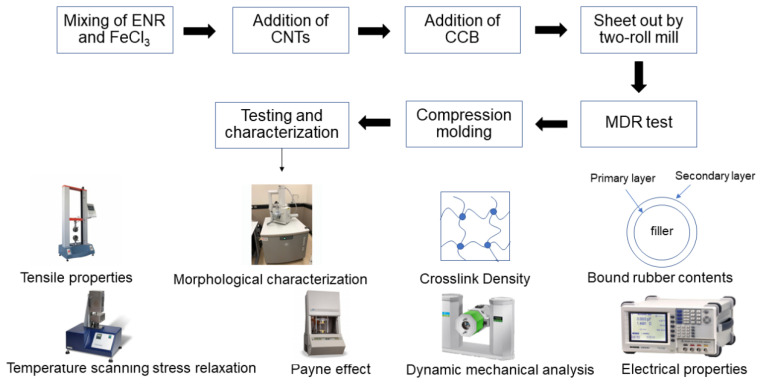
The compounding and characterization procedures of ENR–FeCl_3_/CNT–CCB hybrid nanocomposites.

**Figure 2 polymers-14-04392-f002:**
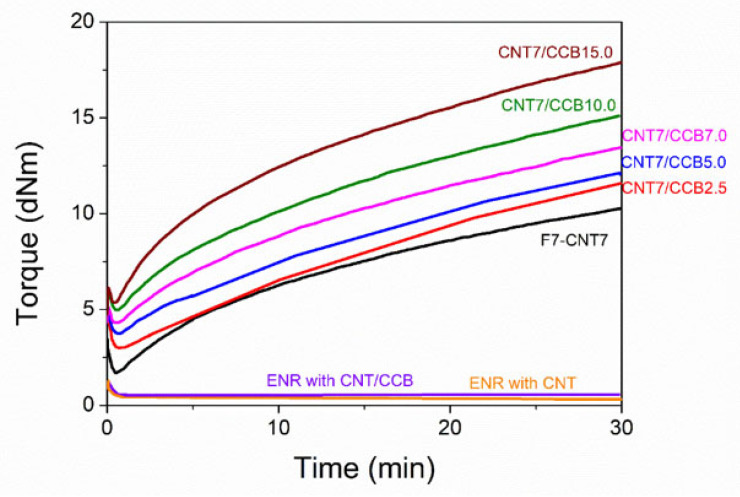
Cure curves of ENR filled with CNTs and CNT–CCB, together with ENR–FeCl_3_ filled with 7 phr of CNT (F7-CNT7) and with 7 phr CNT and various CCB loadings at 2.5, 5.0, 7.0, 10.0 and 15.0 (i.e., CNT7/CCB2.5, CNT7/CCB5.0, CNT7/CCB7.0, CNT7/CCB10.0 and CNT7/CCB15.0).

**Figure 3 polymers-14-04392-f003:**
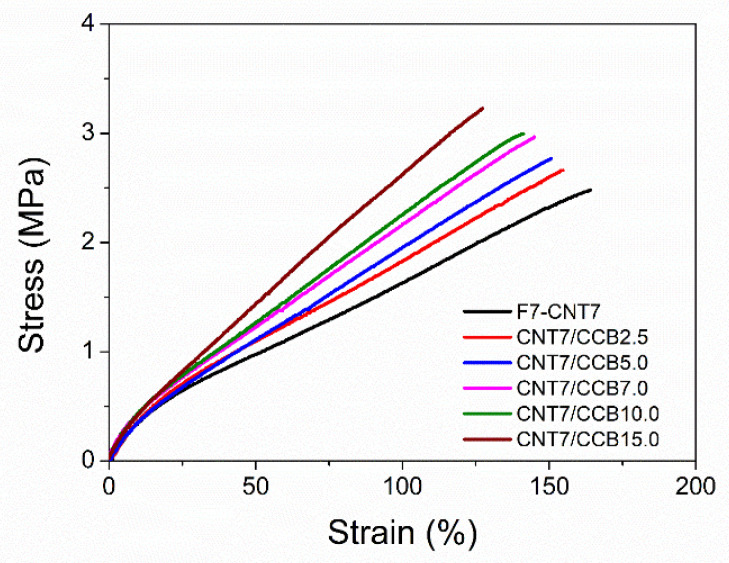
Stress–strain curves of ENR–FeCl_3_ filled with 7 phr of CNT (F7-CNT7) and with 7 phr CNT and various CCB loadings at 2.5, 5.0, 7.0, 10.0 and 15.0 (i.e., CNT7/CCB2.5, CNT7/CCB5.0, CNT7/CCB7.0, CNT7/CCB10.0 and CNT7/CCB15.0).

**Figure 4 polymers-14-04392-f004:**
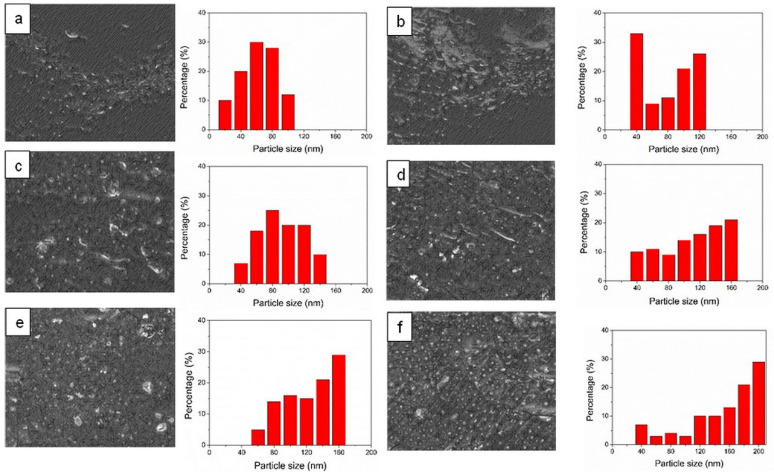
SEM micrographs and average filler particle size of ENR–FeCl_3_ filled with CNTs (F7-CNT7) (**a**) and CNT–CCB hybrid filler with various CCB loadings at 2.5 (**b**), 5.0 (**c**), 7.0 (**d**), 10.0 (**e**) and 15.0 phr (**f**).

**Figure 5 polymers-14-04392-f005:**
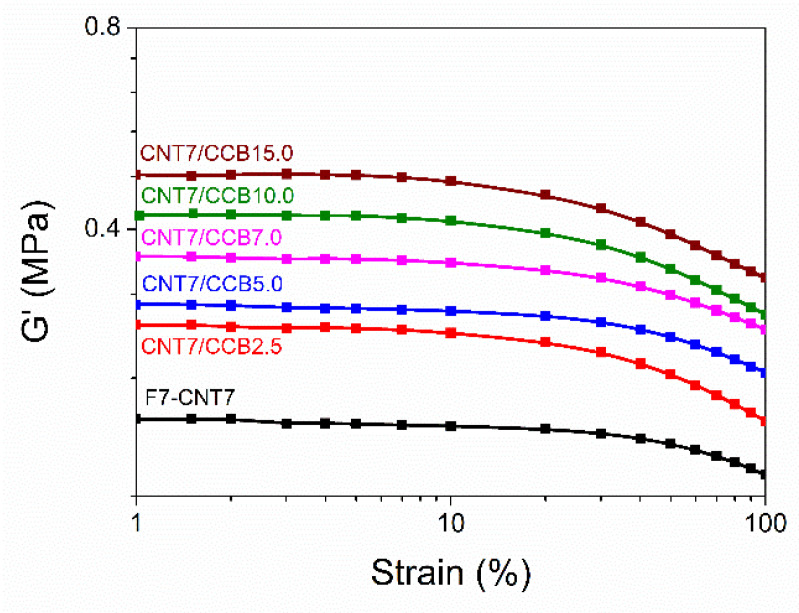
Storage modulus as a function of strain amplitude of ENR–FeCl_3_ filled with CNTs (F7-CNT7) and CNT–CCB hybrid fillers with various CCB loadings at 2.5, 5.0, 7.0, 10.0 and 15.0 phr.

**Figure 6 polymers-14-04392-f006:**
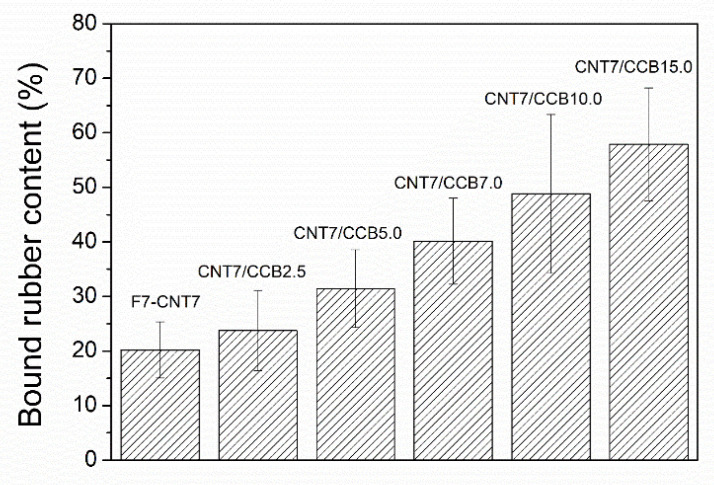
Bound rubber contents of ENR–FeCl_3_ filled with CNTs (F7-CNT7) and CNT–CCB hybrid filler with various CCB loadings at 2.5, 5.0, 7.0, 10.0 and 15.0 phr.

**Figure 7 polymers-14-04392-f007:**
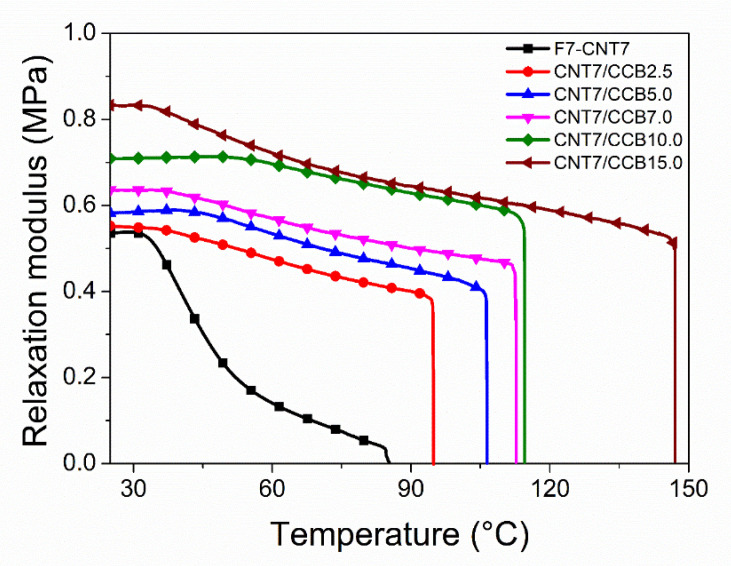
Relaxation modulus as a function of temperature of ENR–FeCl_3_ filled with CNTs (F7-CNT7) and CNT–CCB hybrid filler with various CCB loadings at 2.5, 5.0, 7.0, 10.0 and 15.0 phr.

**Figure 8 polymers-14-04392-f008:**
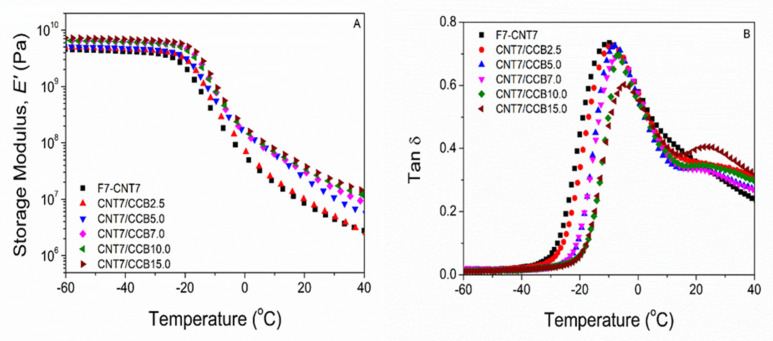
Storage modulus (**A**) and tan δ (**B**) as functions of temperature of ENR-FeCl_3_ filled with CNTs (F7-CNT7) and CNT-CCB hybrid fillers with various CCB loadings at 2.5, 5.0, 7.0, 10.0 and 15.0 phr.

**Figure 9 polymers-14-04392-f009:**
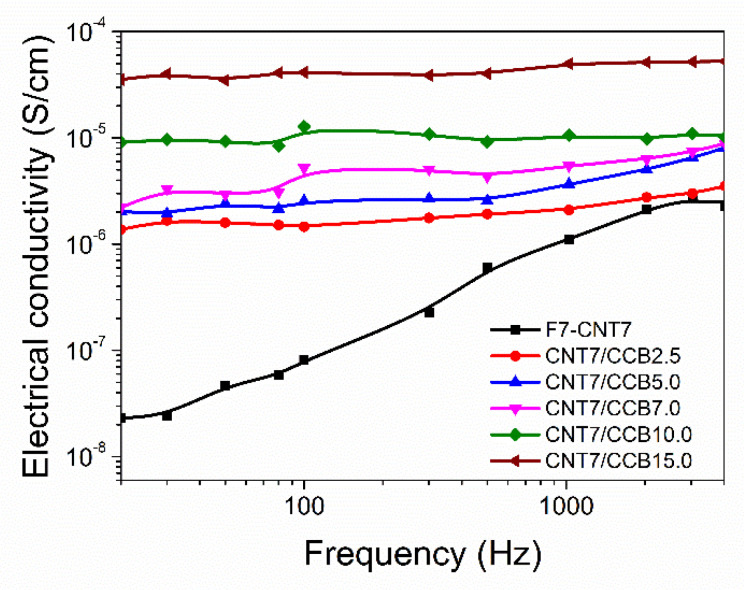
Electrical conductivity as a function of frequency of ENR-FeCl_3_ filled with CNTs (F7-CNT7) and CNT-CCB hybrid filler with various CCB loadings at 2.5, 5.0, 7.0, 10.0 and 15.0 phr.

**Figure 10 polymers-14-04392-f010:**
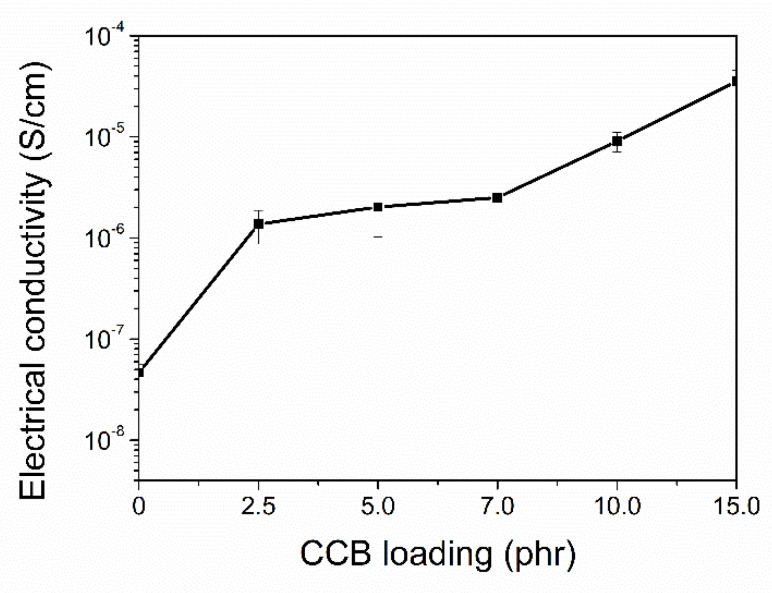
Electrical conductivity (at a frequency at 50 Hz) as a function of CCB loadings of ENR-FeCl_3_ filled with CNTs (0 phr) and CNT-CCB hybrid fillers with various CCB loadings at 2.5, 5.0, 7.0, 10.0 and 15.0 phr.

**Figure 11 polymers-14-04392-f011:**
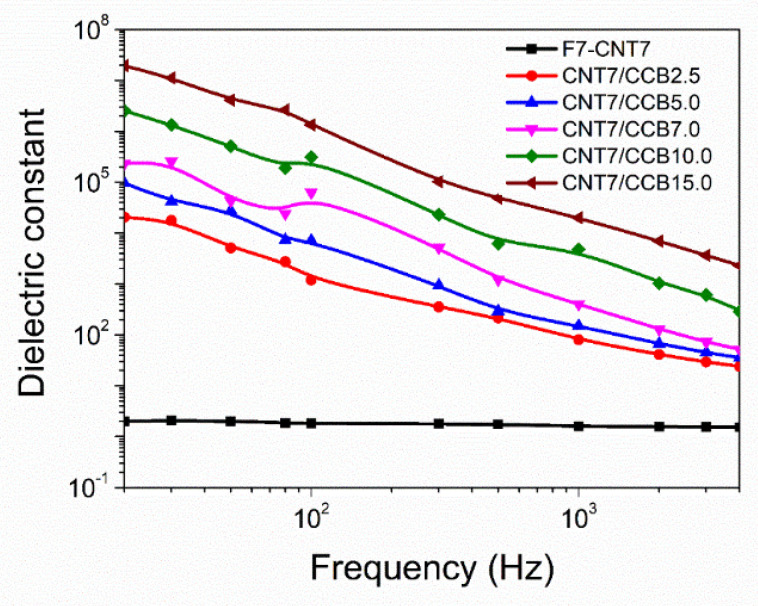
Dielectric constant as a function of frequency of ENR-FeCl_3_ filled with CNTs (F7-CNT7) and CNT-CCB hybrid fillers with various CCB loadings at 2.5, 5.0, 7.0, 10.0 and 15.0 phr.

**Table 1 polymers-14-04392-t001:** Chemical ingredients with sources and compounding formulation of ENR-50 with FeCl_3_ and CNT–CCB hybrid filler.

Chemicals	Sources	Content (phr)
Epoxidized natural rubber (ENR-50)	Muangmai Guthrie Co. Ltd. (Surat Thani, Thailand)	100
Ferric chloride (FeCl_3_)	Sigma-Aldrich Pte Ltd. (Darmstadt, Germany)	7 (mmol)
Carbon nanotubes (CNTs)	Nanocyl S.A. (Sambreville, Belgium)	7
Conductive carbon black (CCB)	Cabot Corporation. (Pampa, TX, USA)	0, 2.5, 5, 7, 10 and 15

**Table 2 polymers-14-04392-t002:** Cure characteristics in terms of minimum torque (*M_L_*), maximum torque (*M_H_*), torque difference (*M_H_-M_L_*), scorch time (*t_s_*_2_) and cure time (*t_c_*_90_) of ENR-50 compounded with FeCl_3_ (ENR–FeCl_3_) and filled with CNTs (F7-CNT7) and CNT–CCB hybrid fillers at various CCB loadings.

Compounds	*M_L_* (dN.m)	*M_H_* (dN.M)	*M_H_-M_L_* (dN.m)	*t_s_*_2_ (min)	*t_c_*_90_ (min)	CRI
F7-CNT7	1.62	9.21	7.59	1.32	5.91	21.78
CNT7/CCB2.5	2.61	11.17	8.56	1.30	5.74	22.52
CNT7/CCB5.0	3.15	11.81	8.66	1.19	5.54	22.98
CNT7/CCB7.0	3.81	12.87	9.06	1.09	5.01	25.51
CNT7/CCB10.0	4.84	14.51	9.67	1.01	4.76	27.32
CNT7/CCB15.0	5.02	17.79	12.77	0.78	4.23	28.98

**Table 3 polymers-14-04392-t003:** Tensile properties in terms of 100% modulus, tensile strength and elongation at break of ENR-50 compounds with FeCl_3_ (ENR–FeCl_3_) filled with CNTs (F7-CNT7) and CNT–CCB hybrid fillers with various CCB loadings.

Materials	100% Modulus (MPa)	Tensile Strength (MPa)	Elongation at Break (%)
F7-CNT7	1.63 ± 0.10	2.58 ± 0.09	165.55 ± 19.11
CNT7/CCB2.5	1.68 ± 0.03	2.67 ± 0.12	155.62 ± 12.13
CNT7/CCB5.0	1.96 ± 0.05	2.83 ± 0.10	144.02 ± 12.22
CNT7/CCB7.0	2.16 ± 0.12	2.96 ± 0.09	141.53 ± 10.31
CNT7/CCB10.0	2.21 ± 0.02	3.16 ± 0.05	137.66 ± 13.77
CNT7/CCB15.0	2.66 ± 0.10	3.43 ± 0.16	136.05 ± 12.89

**Table 4 polymers-14-04392-t004:** Payne effect of ENR-50 compounded with FeCl_3_ (ENR–FeCl_3_) filled with CNTs (F7-CNT7) and CNT–CCB hybrid fillers with various loadings of CCB.

Samples	Payne Effect (%)
F7-CNT7	20.12
CNT7/CCB2.5	31.35
CNT7/CCB5.0	35.12
CNT7/CCB7.0	39.51
CNT7/CCB10.0	42.07
CNT7/CCB15.0	46.41

**Table 5 polymers-14-04392-t005:** Crosslink densities of ENR–FeCl_3_ and its filled composites with CNTs (F7-CNT7) and CNT–CCB hybrid fillers with various loadings of CCB.

Sample	Mooney–Rivlin Eq.	TSSR Evaluation
	Crosslink Densities (mol/m^3^)	Crosslink Densities (mol/m^3^)
ENR–FeCl_3_	110.13 ± 2.01	66.28
F7-CNT7	167.97 ± 2.16	71.71
CNT7/CCB2.5	171.52 ± 8.18	76.98
CNT7/CCB5.0	184.21 ± 1.12	85.27
CNT7/CCB7.0	193.49 ± 1.98	87.52
CNT7/CCB10.0	215.24 ± 10.15	91.65
CNT7/CCB15.0	217.93 ± 10.01	96.55

**Table 6 polymers-14-04392-t006:** Glass transition temperature (*T_g_*) of ENR-50 compounded with FeCl_3_ and (ENR–FeCl_3_) mixed with CNTs (F7-CNT7) and CNT–CCB hybrid fillers.

Sample	Glass Transition Temperature (°C)
F7-CNT7	−18.30
CNT7/CCB2.5	−16.35
CNT7/CCB5.0	−11.31
CNT7/CCB7.0	−10.63
CNT7/CCB10.0	−7.77
CNT7/CCB15.0	−7.68

## Data Availability

Not applicable.

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
