# Peer review of "Ferric Ions Crosslinked Epoxidized Natural Rubber Filled with Carbon Nanotubes and Conductive Carbon Black Hybrid Fillers"

_polymers, 2022, doi:10.3390/polym14204392_

Round 1

Reviewer 1 Report

Dear Editor,

 I have read the manuscript entitled: “Ferric Ions Crosslinked Epoxidized Natural Rubber Filled with Carbon Nanotubes and Conductive Carbon Black Hybrid Fillers: Cure Characteristics, Mechanical, Morphological, Thermal and Electrical Properties” and I would like to address following suggestions to the authors:

1-Please add some lines to indicate the novelty of your study, compare the results with that of the literature and emphasize the novelty of this study.

2-For each research method, it is necessary to expand the discussion. Please add schematic diagram for this study.

3-Line 354. Please put histogram for SEM images.

4-The introduction can be improved by providing a more critical discussion of recent related literature and use new papers. Discuss the shortcomings of previous work and the gaps and how this work intends to fill those gaps. For example, some papers related to electrical properties such as: Materials Science in Semiconductor Processing, 36, 134-139, (2015); Superlattices and Microstructures, 145, 106603, (2020).

5-In the conclusion, the performance findings of the research should have been summarized the innovations and future scope of the work should be highlighted more. 

Author Response

1-Please add some lines to indicate the novelty of your study, compare the results with that of the literature and emphasize the novelty of this study.

Response:  We have added the sentence “The novelty of the current work was the first time to study influence of CNTs-CCB hybrid fillers for reinforcement of epoxidized natural rubber (ENR) crosslinked by coordination bonds via Fe 3+ linkages.” In the text which indicated in blue.

2-For each research method, it is necessary to expand the discussion. Please add schematic diagram for this study.

Response: Thank you very much. We have now added the schematic diagram for the experimental part in this study, in Figure 1 and the blue text.

3-Line 354. Please put histogram for SEM images.

Response: We did not record the histogram of the SEM images. Apologize for this inconvenient.

4-The introduction can be improved by providing a more critical discussion of recent related literature and use new papers. Discuss the shortcomings of previous work and the gaps and how this work intends to fill those gaps. For example, some papers related to electrical properties such as: Materials Science in Semiconductor Processing, 36, 134-139, (2015); Superlattices and Microstructures, 145, 106603, (2020).

Response: I hardly find the relation between these two papers and the contents in our manuscript. Please kindly give more details of your suggestion for us to tackle this point of discussion.

5-In the conclusion, the performance findings of the research should have been summarized the innovations and future scope of the work should be highlighted more.

Response: We have added the new sentence in the conclusion as “Finally, the novel ENR/CNTs-CCB hybrid nanocomposites with coordination cross-linking system and good performance in terms of mechanical a long with electrical properties will encourage a further industrial and research in the field of smart materials.”

Reviewer 2 Report

This manuscript investigated the properties of crosslinked epoxidized natural rubber filled with 3 carbon nanotubes and conductive carbon black hybrid fillers. Most of the Figures and results are grate and reasonable. My comments are as follows

1. L51, 0.05–0.10% (w/w) What is W/W ? maybe wt%

2. What is Øp in Eq.2? Seems p is not in subscript in equation 2.  

3. max and min seem not in subscript on Eq. 6.

4. What is Rp in Eq. 7

5. Figure 1 captions should be below the Figure 1.

6. L352, it well dispersion -> grammar problem

7. Please leave a space between number and unit.

8 How many conductivity tests per CCB loading you have done on Figure 9? It needs standard deviation if test is more than one.

Author Response

  1. L51, 0.05–0.10% (w/w) What is W/W ? maybe wt%

Response: Thank you very much for the referee’s suggestion. L51 has been modified to wt%.

  1. What is Øpin Eq.2? Seems p is not in subscript in equation 2.   ∅

Response: We have now modified  in equation 2, as referee’s comment with thanks.

  1. max and min seem not in subscript on Eq. 6.

Response: Response: We have now modified  in equation 6, as referee’s comment with thanks.

  1. What is Rp in Eq. 7

Response: We have added “Rp is resistance and Cp  is capacitance” in Eq7.

  1. Figure 1 captions should be below the Figure 1.

Response: Thank you very much for the referee’s suggestion. The caption of figure 1 (now is Figure 2) has been revised with thanks.

  1. L352, it well dispersion -> grammar problem

Response: L352 (now 354), it well dispersion have been now modified to “it’s good dispersion”

  1. Please leave a space between number and unit.

Response: Thank you very much for the referee’s suggestion. The space between number and unit has been revised with thanks.

8 How many conductivity tests per CCB loading you have done on Figure 9? It needs standard deviation if test is more than one.

Response: We did 3 replicates with standard deviation now in Figure 9 (now is Figure 10). Thank you very much for the referee’s suggestion.

Reviewer 3 Report

1. The title should be simplified because it is too long.

2. The abstract should be revised because it was short of key data.

3. In Table 3, the mechanical properties should be added error bars.

4. In Fig. 3, I do not agree that the fillers had even dispersion in ENR matrix. I  suggest  the author provide some clear SEM images.

5. In Table 5, "-FeCl3" was wrong, Please correct the formula.

Author Response

  1. The title should be simplified because it is too long.

Response: Thank you very much for the referee’s suggestion. The title has been simplified to “Ferric Ions Crosslinked Epoxidized Natural Rubber Filled with Carbon Nanotubes and Conductive Carbon Black Hybrid Fillers”

  1. The abstract should be revised because it was short of key data.

Response: we have added more sentences in the abstract, as the referee’s suggestion.  

  1. In Table 3, the mechanical properties should be added error bars.

Response: Thank you very much for the referee’s suggestion. The error bars have been added in Table 3.

  1. In Fig. 3, I do not agree that the fillers had even dispersion in ENR matrix. I  suggest  the author provide some clear SEM images.

Response: Thank you very much for the referee’s comment. The SEM micrographs in Figure 3 (now Figure 4) has been readjusted their contrast to have a clearer observation. Also, we have modified the related texts as “It can be seen that the F7-CNT7 shows uneven dispersion of CNTs bundles and some agglomerates in the ENR matrix (Figure 4(a)).”, as seen in the blue texts.

  1. In Table 5, "-FeCl3" was wrong, Please correct the formula.

Response: "-FeCl3" in Table 5 has been revised with thanks.

Round 2

Author Response

 1-Please add some lines to indicate the novelty of your study, compare the results with that of the literature and emphasize the novelty of this study.

Response: We have added the sentence “The novelty of the current work was the first time to study influence of CNTs-CCB hybrid fillers for reinforcement of epoxidized natural rubber (ENR) crosslinked by coordination bonds via Fe 3+ linkages.” In the text which indicated in blue.

Please more explain about choose CCB hybrid.

2nd response: “Furthermore, CCB in the hybrid filler system provided novelty new form filler network of CNTs by connecting the CNTs bundles, promoting superior electrical and mechanical properties of ENR nanocomposites.” In the blue text.

2-For each research method, it is necessary to expand the discussion. Please add schematic diagram for this study.

Response: Thank you very much. We have now added the schematic diagram for the experimental part in this study, in Figure 1 and the blue text.

Please add some related pictures to the schematic diagram.

2nd response: we did as advice.

3-Line 354. Please put histogram for SEM images.

Response: We did not record the histogram of the SEM images. Apologize for this inconvenient.

No need to record histogram. Please use related software to draw histogram.

2nd response: we did as advice.

4-The introduction can be improved by providing a more critical discussion of recent related literature and use new papers. Discuss the shortcomings of previous work and the gaps and how this work intends to fill those gaps. For example, some papers related to electrical properties such as: Materials Science in Semiconductor Processing, 36, 134-139, (2015); Superlattices and Microstructures, 145, 106603, (2020).

Response: I did not find any relation of these two papers with the contents in our manuscript. Please kindly give more details of your suggestion for us to tackle this point of discussion.

As you mention in line 87-88 graphene and also electrical properties (related your study and mention in figure1), please more explain about properties of them in introduction.

2nd response: in line 87-88  “Various types of hybrid fillers have been investigated in CNTs filled ENR and epoxy containing polymer system including carbon black [30, 31], nano-clay (NC) [32], graphene nano platelets [33], zinc oxide [34], silver nanoparticles (AgNPs) [35] and conductive carbon black (CCB) [36].”  It is obvious that we described the hybrid composite of CNTs filled epoxy containing polymer like ENR and epoxy resin only. The graphene was one of which was used in the hybrid CNTs-graphene system in Ref 33.  We are sorry to inform the referee that we still could not add the recommend papers into the contents in our introduction due to lack of close relation to the main contents of  our work. Thank you very much for your kind suggestion.
